

# Inclusion of biochar in a C-dynamics model based on observations from a 8 years field experiment

Roberta Pulcher[1], Enrico Balugani[2], Maurizio Ventura[3], Diego Marazza[1,2]

[1]Department of Biological, Geological and Environmental Sciences, BIGeA, Università di Bologna, Bologna, Italy
[2]Department of Physics and Astronomy, Università di Bologna, Bologna, Italy
[3]Faculty of Science and Technology, Libera Università di Bolzano, 39100 Bolzano/Bozen, Italy

*Correspondence to*: Enrico Balugani (enrico.balugani2@unibo.it)

**Abstract.** Biochar production and application as soil amendment is a promising carbon (C) negative technology to increase soil C sequestration and mitigate climate change. However, there is a lack of knowledge about biochar degradation rate in 10 soil and its effects on native soil organic carbon (SOC), mainly due to the absence of long term experiments performed in field conditions. The aim of this work was to investigate the long term degradation rate of biochar in a field experiment of 8 years in a poplar short rotation coppice plantation in Piedmont (Italy), and to modify the RothC model to assess and predict how biochar influences soil C dynamics. The RothC model was modified by including two biochar pools, labile (4% of the total biochar mass) and recalcitrant (96%), and the priming effect of biochar on SOC. The model was calibrated and 15 validated using data from the field experiment. The results confirm that biochar degradation can be faster in field conditions in comparison to laboratory experiments; nevertheless, it can contribute to substantially increase the soil C stock in the long-term. Moreover, this study shows that the modified RothC model was able to simulate the dynamics of biochar and SOC degradation in soils in field conditions in the long term, at least in the specific conditions examined.

## 1 Introduction

Biochar, the solid product of pyrolysis or gasification of biomass, has a large potential for increasing soil carbon (C) stocks and improve soil quality worldwide (Woolf et al. 2010; Smith 2016; European Academies' Science Advisory Council 2018). Due to its stability and resistance to mineralization, adding biochar to soil is considered a viable strategy for climate change mitigation (Lehmann et al. 2006; Zahida et al. 2017), since it can increase soil C stocks for hundreds or thousands of years (Wang et al. 2016). Among the negative emission strategies proposed by IPCC (2014), biochar has the lowest impact in 25 terms of water footprint, land use and costs (Smith 2016).

In order to assess the potential of biochar for soil C sequestration (SCS), two things are required: long term experimental data of biochar degradation in field conditions and a working model of the degradation of biochar in soils, in order to generalize and upscale experimental findings. In fact, the generalisation of results from experiments is not easy, since biochar degradation depends on several factors such as: biochar characteristics, e.g. the original feedstock and the production 30 temperature (Cetin et al. 2005a; Saffari et al. 2020b; Ippolito et al. 2020a); the characteristics of the soil to which biochar is





applied, such as its clay and mineral content; the conditions affecting the soil biochar interaction, like the climate and the vegetation in the area (Wang et al. 2016; Han et al. 2020). Even though many studies have been performed on biochar degradation in soils, a wide range of degradation rates have been estimated, mainly from laboratory studies, resulting in large uncertainties on biochar stability (Luo et al. 2011; Fang et al. 2013; Han et al. 2020).

Biochar mineralization to $CO_2$ is mostly described with a double exponential decay model (Zimmerman et al. 2011c), according to which biochar is composed by two fractions with different degradation rates: a labile fraction with a larger degradation rate and a recalcitrant fraction with a smaller degradation rate. The labile fraction constitutes usually about 2-5% of the total mass, but can rise up to 20% depending on the feedstock used (Cetin et al. 2005b; Han et al. 2020; Ippolito et al. 2020b; Saffari et al. 2020a), and can be directly mineralized by the soil microbial community. The recalcitrant part makes up

the rest of the biochar mass (Wang et al. 2016) and is usually regarded as resistant to direct microbial oxidation (Guo and Chen 2014; Ippolito et al. 2020a). This simple, empirical model, however, does not take into account the processes that lead to biochar degradation, and cannot be generalized to different climates and soil types, which can only be included in more complex models describing C dynamics in soil.

Biochar addition to soil affects the organic C storage not only by directly increasing the amount of soil C, but also indirectly

influencing the turnover of native soil organic C (SOC), a phenomenon referred to as priming effect (Kuzyakov et al. 2014; Maestrini et al. 2015). Biochar has been found to increase (positive priming effect), decrease (negative priming effect) or have no effect on soil organic matter degradation (Lehmann and Joseph 2015), according to soil and biochar type and experimental conditions and duration (Zimmerman et al. 2011a). The priming effect has been reported to change over time: fresh biochar can have a negative priming effect at the beginning, while the same, aged biochar can have a positive priming

effect later on (Jiang et al. 2019). The determination of priming effect of biochar on SOC is fundamental to determine the C sequestration potential of biochar (Gurwick et al. 2013).

The degradation rate of biochar and its interaction with SOC are usually estimated through laboratory incubation studies (Leng et al. 2019b), most of them are short term (Cross and Sohi 2011; Bruun et al. 2014) and only a few last for years (Kuzyakov et al. 2014). However, laboratory studies may not be representative of complex environmental conditions since

they may miss some important processes due to their controlled conditions (Ventura et al. 2015a). Therefore, field experiments are essential to understand the dynamics of biochar in soils. Yet, despite their importance, field scale experimental studies on soil biochar degradation are still scarce (Jones et al. 2012; Gurwick et al. 2013).

Another relevant aspect affecting the calculation of biochar degradation rates and its effect on soil organic matter, both in laboratory and field experiments, is the duration of the study (Chao et al. 2018). It has been observed that short term trials

result in lower biochar mean residence time (Leng et al. 2019a), due to the decomposition of the labile biochar fraction, and an overestimation of the positive priming effect, which normally occurs in the first few months. It has been proposed that field studies to determine SOC degradation rate should have a duration of about 10 years, in order to detect changes and temporal shifts in trends, and that long term datasets should be introduced in established and new models to test their



performance (Smith et al. 2020). Therefore, the results from medium and long term trials are fundamental to assess the SCS potential of biochar (Ventura et al. 2019a).

Models are widely used to generalize SOC dynamic studies and to extend their findings in space (e.g., obtaining SOC maps for wide areas; Farina et al. 2018) and/or in time (e.g., projecting SOC changes in soils to the future, with respect to some soil management changes; Meyer et al. 2018). One of the most well-known and widely used models for soil C dynamics is the RothC model (Coleman and Jenkinson 1996a). The reason for the success of RothC is that it is a simple model and it requires relatively few and easily obtainable parameters and input data about vegetation management, soil and climate characteristics.

To date, only limited attempts have been made to include biochar degradation in SOC dynamic models. Mondini et al. (2017) modified the RothC model to simulate the mineralization of exogenous organic matter, but without specific model representation for biochar. Lefebvre et al. (2020) developed a biochar submodel for the RothC, but they did not calibrate nor validate it with experimental data. Overall, existing models of biochar degradation in soils rely on literature data deriving from laboratory or short term studies and have not been calibrated or validated in dedicated experiments.

The main objective of this study was to model the degradation of biochar in field conditions, using a modified version of RothC. Therefore, we: (a) modified the RothC model to include biochar; (b) calibrated and validated the modified RothC using data from a study performed in a short rotation coppice plantation over an eight year period (Ventura et al. 2019a). Our specific objectives were:

- To modify the RothC model to include biochar as a carbon pool, and its priming effect on SOM.
- To determine the biochar degradation rate under field experimental conditions in the long term.
- To verify the priming effect of biochar on SOM in the long term. In particular, we aimed to assess whether the negative priming effect of biochar previously observed (Ventura et al. 2015b) remained the same or not.

## 2 Materials and methods

### 2.1 Experimental data

#### 2.1.1 Study area

The experimental field is a short rotation coppice (SRC) plantation of poplars (*Populus x Canadensis Mönch*, Oudemberg genotype), located in Prato Sesia (Novara) in northern Italy (45° 390 32.2812″ N; 8° 210 16.8339″ E; Ventura et al. 2015b; Ventura et al. 2019a). Since its establishment in 2010, the SRC plantation has never been irrigated nor fertilized. The field is arranged in single rows with three m wide interrow and a plant density of 6600 trees ha$^{-1}$. During the experimental period between 2012 and 2014, trees were harvested twice: in March 2012, before biochar application, and two years later, in March 2014. After 2014, no other cuts have been performed.



The soil is an Entisol, according to the USDA classification, with a sandy loam texture (12% clay, 34% silt, and 54% sand).
Soil pH is 5.4 (in water), total soil N content is 0.11% and SOC content is 1.4%. The climate in the study area is classified as temperate with warm summer (Kottek et al. 2006). Daily averages of air temperature range between 0°C and 25°C, with an annual average of 12 °C; precipitations, with annual average of 1500 mm, increase in intensity during spring and autumn, reaching peaks of 300 mm month$^{-1}$. Relative humidity is on average 51%. Data were obtained from the environmental regional agency 'Arpa Piemonte'.

**2.1.2 Data and flux measurements for model calibration**

Experimental data used to calibrate the model were collected within the EU-FP7 EuroChar project, with the aim to determine biochar stability and priming effect on SOC over a three years period, from 2012 until 2014 (Ventura et al. 2015b; Ventura et al. 2019a). The biochar used was produced from maize (*Zea mays* L.) silage feedstock pellets by gasification at 1200°C, at atmospheric pressure, with a residence time of 40 min in the gasification plant (©A.G.T. – Advanced Gasification
Technology s.r.l., Cremona, Italy). The isotopic signature corresponds to –13.8‰, the H:C atomic ratio is 0.5 and C:N is 42.9. Further information on biochar physicochemical characteristics can be found in Ventura et al. (2015b).

A completely randomized experimental design was used, with four biochar treated plots and four control plots. Plots (45 m$^2$ each) included three rows of nine plants each. On 30 March 2012, maize biochar (30 t C ha$^{-1}$, corresponding to 16.8 t C ha$^{-1}$) was incorporated into the first 15 cm soil layer of the short rotation coppice by rotary hoeing. Hoeing was also carried out in
control plots, to disturb soil as in biochar treated plots.

Monthly aboveground C inputs due to poplar leaf litter was measured directly on the experimental site using rectangular litter traps, set up in the field in August 2012, to cover a representative area (from the central row to the middle of the interrow) of the central interrow of each plot, as described in Ventura et al. (2019b). Litter was collected from the traps monthly, from September 2012 to November 2014. The collected litter was dried at 65°C in an oven, weighted and analysed
to determine its C content as described for soil samples. As the poplar plants were not cut anymore after March 2014, the litter C inputs from year 2015 to 2020 were calculated using the litter production in the second year after the first cut (i.e., 2013), which were considered more similar to the litter production in the following years.

In six plots (three per treatment), trenched subplots (50x50 cm) were set up, one per each plot, between two plant rows, by digging 60 cm deep and 15 cm wide trenches. A geotextile canvas (Typar®, Dupont, Wilmington, DE, USA) was inserted in
the trenches to isolate each subplot from root ingrowth, allowing gas and water exchange. On each subplot, soil $CO_2$ efflux (heterotrophic respiration, due to soil microbes and soil fauna) was measured using an automatic soil respiration system connected to closed automated chambers. Soil water content (SWC) between 0 and 18 cm depth and soil temperature at 10 cm were recorded every 30 minutes using water content reflectometers (CS-616, Campbell Scientific, Logan UT, USA) and temperature probes (107, Campbell Scientific, Logan UT, USA), respectively. Total and heterotrophic respiration
measurements were averaged on daily basis and gaps in the database were filled using the model proposed by Qi & Xu (2001):



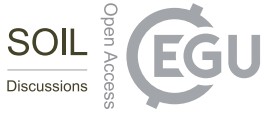

$$R = p_1 T^{p_2} \cdot SWC^{p_3}, \qquad\qquad (1)$$

where R is the soil $CO_2$ efflux (total or heterotrophic), T is the soil temperature (°C), SWC is the soil water content (%) and $p_1$, $p_2$ and $p_3$ are empirical parameters.

Soil sampling was carried out in January 2013 and in March 2015, in each of the four plots per treatment, to determine the total SOC content and the remaining biochar C stock in soil to a depth of 40 cm (Ventura et al. 2019a). Maize biochar has an isotopic signature ($\delta^{13}C_{biochar} = -13.8‰$) distinguishable from that of the SOC in the plantation ($\delta^{13}C_{SOC} = -23.5‰$). This allowed to calculate the amount of $CO_2$ deriving from biochar degradation and from native SOC degradation, using an isotopic mass balance. The amount of biochar remaining at the end of the experiment, which was 86 and 79% of the original

amount in the absence and in the presence of plant roots, respectively, was estimated by subtracting the C amount decomposed to $CO_2$ from the C amount initially added to soil with biochar application. The degradation rates of biochar were assessed using a double exponential decay model (Ventura et al. 2019a): in presence of plant roots, the degradation rate of the more recalcitrant biochar fraction (96% of the total) was $k_1 = 0.08$ y$^{-1}$ and the degradation rate of the labile biochar fraction (4%) was $k_2 = 2.55$ y$^{-1}$. Since environmental conditions can change (Jiang et al. 2019), we decided to go back on the

field after 8 years and measure the amount of biochar and SOC with the same methodology of Ventura et al. (2019a).

**2.1.3 Determination of SOC and biochar-C stock for model validation**

In October 2020 another soil sampling was performed in the plantation, aimed at quantifying SOC and biochar stocks in soil after 8 years from the beginning of the experiment. In summer 2020, the construction of a methane pipeline affected the experimental field, disturbing the soil in four out of the eight original plots. Therefore, the sampling was limited to the four

remaining plots (two biochar treated and two control plots). From each of these plots, 10 soil samples were collected, using a 2.5 cm diameter auger (Eijkelkamp, Giesbeek, The Netherlands), at 0-20 cm and 20-40 cm depth, for a total of 80 samples. Samples were collected at 0, 37.5, 75, 112.5, and 150 cm from the central poplar row in each plot along two lines perpendicular to the plant row, in correspondence to the third and the sixth plant of the row. Additional soil samples were collected in two points for each plot with a sample ring kit (Eijkelkamp, Giesbeek, The Netherlands) to calculate the soil

bulk density.

The collected samples were sieved at 2 mm, finely ground with a ball mill (Retsch MM 400, Germany) and analysed with a continuous flow isotopic ratio mass spectrometer (CF-IRMS, Delta V Advantage, Thermo Fisher Scientific, Bremen, Germany) for the determination of the content (%) and the isotopic signature ($\delta^{13}C$) of the soil organic C. For each plot, the SOC stock (g C m$^{-2}$) at 0-20 and 20-40 cm depths was calculated as follows (Ventura et al. 2019a) :

$$SOC\ stock = \frac{C_i}{100}\ \rho_{soil}\ d\ , \qquad\qquad (2)$$

where $C_i$ is the organic C content (%) at the considered soil layer, $\rho_{soil}$ is the soil bulk density (g m$^{-3}$), and $d$ is the depth of the soil layer (0.2 or 0.4 m).





The fraction of biochar-C on total SOC ($f_B$) was calculated for each soil layer by an isotopic mass balance, as follows:

$$f_B = \frac{\delta^{13}C_B - \delta^{13}C_C}{\delta^{13}C_{Biochar} - \delta^{13}C_C}, \tag{3}$$

where $\delta^{13}C_B$ and $\delta^{13}C_C$ are the isotopic signatures of the SOC in biochar treated and untreated soils, respectively, and $\delta^{13}C_{Biochar}$ is the isotopic signature of the applied biochar (-13.8 ‰; Ventura et al. 2019a).

Therefore, the biochar-C stocks (g C m$^{-2}$) at 0-20 and 20-40 cm depths was calculated multiplying the SOC stock at each layer for the respective $f_B$ values. The total soil biochar-C stock in the 0-40 cm layer was obtained by summing the amounts obtained in the two layers. The remaining amount of biochar-C, as percentage of the initial amount applied to soil (0-40 cm

depth), was therefore calculated. The native SOC stock (excluding biochar) in biochar treated soil was obtained subtracting the biochar-C stock from the total SOC stock (original SOC + biochar-C) in biochar-treated plots.

Measured total SOC stock in biochar-treated and control plots in the different sampling years was compared using analysis of variance (ANOVA) with biochar and year as factors, including the interaction between the two.

## 2.2 Modelling

### 2.2.1 The RothC model

RothC-26.3 is a model for the turnover of organic C in non-waterlogged topsoils, that accounts for the effects of soil properties, temperature, moisture content, and plant cover on the turnover process. RothC was originally developed with the aim of modelling the organic C turnover of long term field experiments in arable soils in Rothamsted (West Common, UK), then it was adapted to operate in different ecosystems, including croplands, grasslands and forests (Coleman and Jenkinson

1996b; Falloon and Smith 2002). In RothC, SOC is subdivided into four active carbon pools, where carbon decreases as first order exponential decay; the degradation rate constants of the four active compartments used in the model are: $k_{DPM} = 10$ yr$^{-1}$ for DPM, $k_{RPM} = 0.30$ yr$^{-1}$ for RPM, $k_{BIO} = 0.66$ yr$^{-1}$ for BIO, and $k_{HUM} = 0.02$ yr$^{-1}$ for HUM. The degradation rates can be modified by three factors which account for the effect of air temperature (factor *a*), soil moisture (factor *b*) and soil cover (factor *c*) on the mineralization rate of SOC. Carbon inputs enter the soil as either DPM or RPM, and then transform into

HUM, BIO and $CO_2$ with proportions defined by an empirical equation depending on soil clay content (Eq. 4). HUM and BIO also decompose to $CO_2$ at every time step.

$$C\ output\ = C_{POOL}(1 - e^{-a\ b\ c\ k_{POOL}\frac{1}{12}}), \tag{4}$$

Where $C_{POOL}$ indicates the specific C pool (BIO, or HUM) and $k_{POOL}$ is the specific degradation rate for each pool. The monthly C inputs (t C ha$^{-1}$) are defined by the user. The DPM/RPM ratio of the C inputs can be set by the user as well, and is

usually chosen amongst the values suggested by RothC manual (1.4 for most agricultural crops and improved grasslands; Coleman and Jenkinson 1996).



### 2.2.2 Modification of the RothC model: BC-RothC

The standard RothC model was modified to include two C pools, represented by the labile (BClab) and recalcitrant (BCrec) biochar fractions (Fig. 1). From here on the modified model is defined as the BC-RothC model. The initial proportion of

BClab and BCrec, 4% and 96% and their specific degradation rates $k_1$ and $k_2$, were previously estimated by Ventura et al. (2019a) in the same site (Section 2.1.2). Before their introduction in the model, $k_1$ and $k_2$ were divided by the yearly average rate modifying factors $a$, $b$, and $c$, (Eq. 5). In this way, it was possible to extrapolate the decay rates independently from these environmental factors.

$$k_{REC} = \frac{k_1}{a\,b\,c} \; ; \; k_{LAB} = \frac{k_2}{a\,b\,c},$$    (5)

Where $k_{REC}$ is the degradation rate of BCrec in the modified RothC model and $k_{LAB}$ is the degradation rate of BClab in the modified RothC model. Since the field measurements showed no effect of biochar on soil temperature and soil water balance, the equations for $a$, $b$, and $c$ rate modifying were not changed in the BC-RothC model.

It was assumed that biochar in soil partly mineralizes into $CO_2$, partly moves into BIO and HUM pools. As Mondini et al. (2017b), we assumed biochar does not enter DPM and RPM, which are direct C input to soil, but contributes to the more

stable C pools, BIO and HUM. $CO_2$ emitted from biochar treated soil (heterotrophic respiration) is given by the sum of the outputs of $CO_2$ from DPM, RPM, BIO, HUM, BClab and BCrec.

A negative priming effect of biochar on SOC was previously observed in the same site by Ventura et al. (2015b; 2019a), who reported that biochar reduced SOC degradation by 16% each year, over a three year period. From these observations, the priming effect was introduced in the model as a constant ($pe$ factor) reducing SOC turnover by 16%. Consequently, the

equation determining the output of C as $CO_2$ from the different C pools became:

$$C\ output\ = C_{POOL}(1 - e^{-a\ b\ c\ pe\ \text{k}_{POOL}\ \frac{1}{12}})$$    (6)

Where $pe$ is the priming effect factor, i.e., log(0.16).



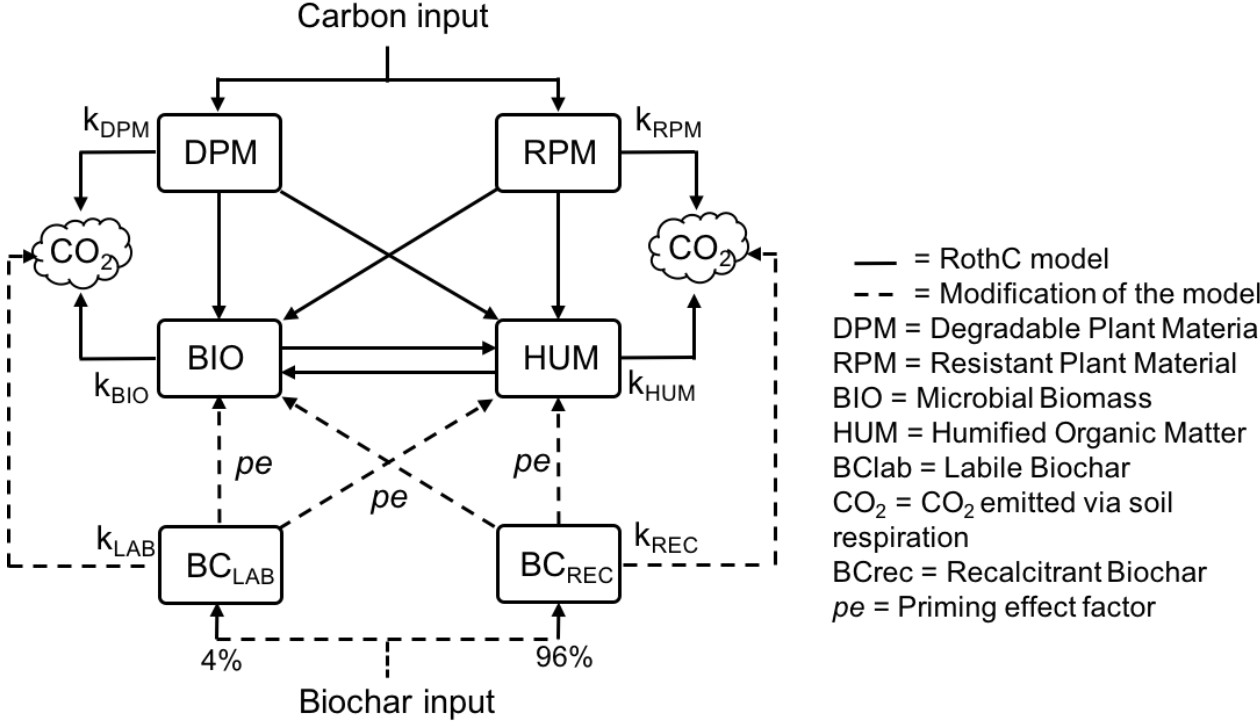

**Figure 1.** Modification of the RothC model with the inclusion of labile and recalcitrant biochar pools, and the priming effect on BIO and HUM. The $CO_2$ pool is assumed to be comparable to soil heterotrophic respiration.

### 2.2.3 Simulations

Modelling was divided in five steps: (1) Initially, the unmodified RothC model (from here on defined as control model) was initialized with a 'spin up run' method (Nemo et al. 2017) to obtain the proportion in which SOC is split among the four active C pools; (2) After the spin run, the control model was calibrated using the experimental data of soil respiration and SOC collected between 2012 and 2014 (Section 2.1.2), with the aim to adjust the parameters that introduce uncertainty in the model; (3) After calibration, the control model was validated against eight years dataset (SOC measurements from 2012 to 2020; Section 2.1.3), to assess the suitability of the unmodified RothC model to represent SOC dynamics in Prato Sesia control plots; (4) The BC-RothC model constants were calibrated using heterotrophic respiration and SOC measurements from biochar treated plots in 2012-2014; (5) After calibration, the BC-RothC model was validated with SOC measurements from 2013 to 2020.

The control model was initialised with a 'spin up run' under the assumption that the soil at the beginning of the experiment was in equilibrium condition: the initial C pools were all set to zero and the model was run under average meteorological conditions and C inputs, until equilibrium was reached (i.e., the average yearly SOC stock did not change significantly



anymore). The relative proportion of the different C pools, estimated by the spin up run at equilibrium conditions, was therefore used to subdivide the total amount of SOC measured in the field in 2013 (Ventura et al. 2019a) into the different RothC pools. Monthly average air temperature and monthly precipitation data recorded in the area from 2001 to 2020, provided by the regional environmental agency, were used to define the meteorological conditions of the "average year" used in the spin up run (climatic input). Data about soil characteristics and agricultural practices needed by the model (soil
input) were obtained by previous studies in the area or estimated using other literature data (Ventura et al. 2015).

For the calibration of the control model, monthly average air temperature and monthly cumulative rainfall in 2012, 2013 and 2014 (climatic inputs), were obtained by the meteorological station installed in the field (Ventura et al. 2019a) and from a nearby meteorological station managed by the regional environmental agency; monthly evapotranspiration in the period was calculated from meteorological data with the FAO Penman Monteith formula (Allen et al. 1998).

Soil depth was set to 40 cm. Aboveground C input from poplar plants was set as measured (see section 2.1.2). Belowground C input from poplar plants was estimated from a study performed in a similar poplar SRC plantation in northern Italy (Ventura et al. 2019c). The monthly above and belowground C input from grass was estimated from studies performed in different areas but in similar conditions (Zanotelli et al. 2013; Pausch and Kuzyakov 2018). The sum of above and belowground C inputs from grass and poplar plants was set as total organic C input to soil in the control model.

The DPM/RPM ratio of the organic C input to soil was calculated as the weighted average of the values for poplar leaves and grass, using their relative contribution to total soil C input as weights. The DPM/RPM ratio for poplar leaves and grass were set to 1.2 and 2.4, respectively, on the base of literature data (Zanotelli et al. 2013; Pausch and Kuzyakov 2018; Ventura et al. 2019c). Monthly soil cover from vegetation was determined according to field observations in the experimental period.

The control model calibration was performed to adjust the values of the C input from grass and the soil clay content, which
were considered the parameters with larger uncertainty. Two datasets were used to calibrate the model: the heterotrophic soil respiration dataset and the SOC stock measurements. The RothC (and BC-RothC) model calculates the outflow of $CO_2$ from the soil due to mineralization of SOM (and biochar), which can be compared with measured heterotrophic respiration ($CO_2$ flux measured in trenched plots). However, the SOC measurements were collected in untrenched plots; therefore, the model was calibrated by running simultaneously simulations of trenched and untrenched plots. The daily heterotrophic respiration
measurements were aggregated to obtain monthly $CO_2$ flux out of the soil and compared with the $CO_2$ monthly production calculated by RothC. The $CO_2$ measurements, due to their high frequency and representativity had a very low uncertainty and are, thus, much better suited to calibrate the RothC model then SOC measurements (Mondini et al. 2017a; Leng et al. 2019a). Thus, one hundred times more weight was given to the respiration data than to the SOC measurements for calibration purposes. The calibration used the Powell optimization method to minimize the difference between measured and
simulated values. The calibration ran for three years and utilized a timestep of 0.0625 d$^{-1}$. Validation of the control model was performed by comparing simulation of SOC trend over eight years against SOC measurements taken in January 2013, March 2015 and October 2020.





The same climate inputs of the control model were used in the BC-RothC model. As soil inputs, the adjusted values of C inputs to soil from grass and soil clay content, resulting from the calibration of the control model, were used. An extra input

of 16 t C ha$^{-1}$ due to biochar application was set at time 0. The initial conditions for the active soil C pools were also the same as those of the control model, since the assumption is the same (previous equilibrium conditions, no previous biochar application). The biochar model calibration was intended to adjust the values of $k_{LAB}$ and $k_{REC}$, which had been determined starting from a simpler double exponential decay with no dependence on weather conditions. As in the case of the control model, the soil respiration and SOC data collected over the same three year period in the biochar treated plots were used for

calibration, and SOC measurements in 2013, 2015 and 2020 were used to validate the prediction of the BC-RothC model in the long term.

## 3 Results

### 3.1 Experimental data: SOC and biochar C stocks

The comparison between SOC stock in control and biochar treated plots across the three years of sampling (January 2013,

March 2015, October 2020) shows that the amount of C in biochar treated plots was always higher than in control plots, as no interaction between biochar and year was found. Biochar-C stock in the 0-20 cm soil layer decreases from 2013 to 2020, due to biochar mineralization. In 2013, 81% of the initial amount of the added biochar was still present in the soil. This amount decreased to 63.3% in 2015 and 60.1% in 2020.

### 3.2 Control model

In the spin run, the equilibrium was reached after approximately 2000 months of simulation (167 years), and the simulated SOC stock was approximately 29 t C ha$^{-1}$, distributed in the different pools according to these proportions: 1.5% DPM, 15% RPM, 2.5% BIO, 81% HUM. Multiplying those percentages by 80.4 t C ha$^{-1}$, which is the SOC stock measured in 2013, the SOC stocks in the four active pools were: DPM = 1.0 t C ha$^{-1}$, RPM = 10.7 t C ha$^{-1}$, BIO = 1.8 t C ha$^{-1}$, HUM = 66.8 t C ha$^{-1}$. The control model simulation showed good agreement with measured soil respiration even before calibration (Fig. 2a). The

calibrated input of C from grass input was 0.124 t C ha$^{-1}$, two times the value set before calibration, while the calibrated clay content was in the upper limit of the measured values (17%). After calibration, the simulated soil respiration fitted closely the measurement data and generally fell within the standard error of the measurements, with the exception of summer period, when downwards peaks were observed in the simulated values but not in the measured ones, particularly in year 2012. The initial SOC content (80 t C ha$^{-1}$) regularly decreased over eight years to 70 t C ha$^{-1}$ (Fig. 2c). At the end of validation in year

2020 the projection of SOC content in control plots (70.3 t C ha$^{-1}$) is close to the measured value (59.1 t C ha$^{-1}$).



### 3.3 BC-RothC model

The BC-RothC model simulation properly fitted soil respiration measurements even before calibration (Fig. 2b). BC-RothC model calibration resulted in a small change in the degradation rates $k_{LAB}$ (from 2.55 to 3.6 $y^{-1}$) and $k_{REC}$ (from 0.08 to 0.14 $y^{-1}$), indicating a slightly faster mineralization of biochar. After calibration, the simulation result fitted the measured data and

generally fell within the error bars, except for the downward summer peaks, in the same way of the control model simulation.

The simulated trend of total SOC (native SOC + biochar-C) stock shows an initial peak, due to the addition of biochar into soil (Fig. 2d). At the end of the simulation (month 105) SOC reaches the value of 79 t C $ha^{-1}$, which is very close to the measured value (72.4 t C $ha^{-1}$). The simulated native SOC (without biochar-C) was 73 t C $ha^{-1}$; thus, after eight years of

simulation, 5 t C $ha^{-1}$ of the total SOC stock in biochar plots are attributable to biochar C. Furthermore, the native SOC stock simulated in biochar plots is about 4 t C $ha^{-1}$ higher, in 2020, than control plots (70.3 t C $ha^{-1}$, Fig 2d). According to the BC-RothC model prediction, the biochar-C amount in soil, eight years after application, is 5.7 t C $ha^{-1}$ (Fig. 2d). This value corresponds to the 54.7% of the initial biochar-C amount, which is about 5% lower than the measured remaining amount (60 ± 14%). Therefore, the BC-RothC model slightly underestimates the remaining biochar.

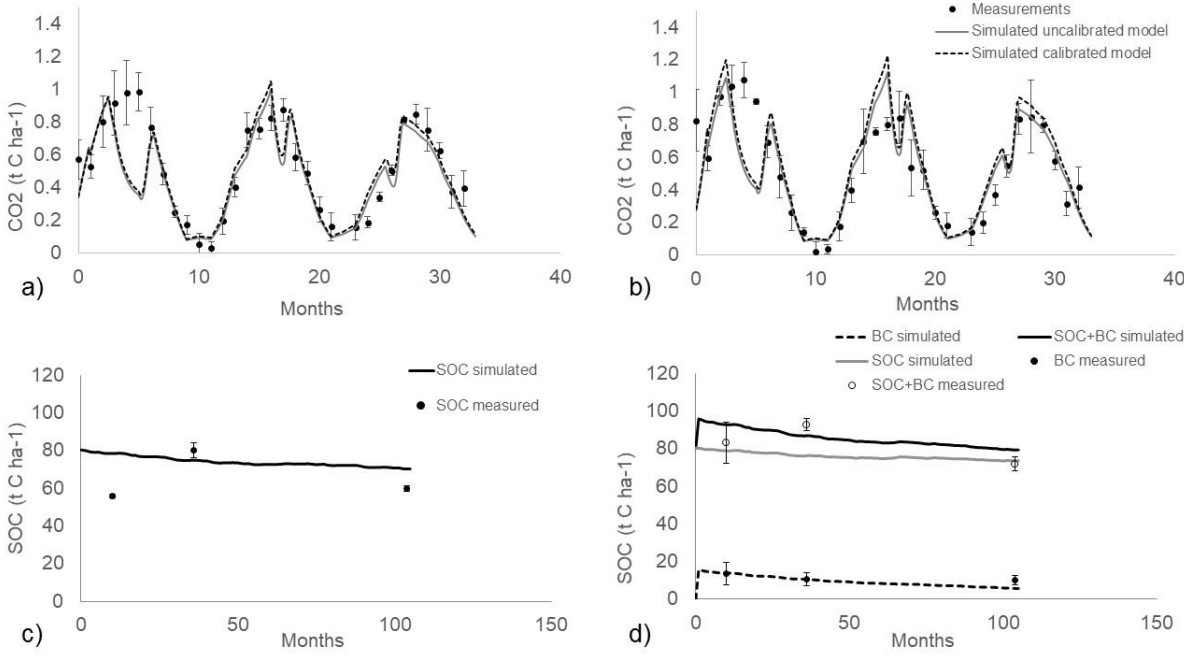


**Figure 2.** (a) Measured (circles) and simulated (lines) heterotrophic respiration, according to the control model before calibration (solid lines) and after calibration (dashed lines), across a three year period (2012-2014). (b) Measured (circles) and simulated heterotrophic respiration, according to the biochar model, before calibration (solid lines) and after calibration (dashed lines), across a three year period (2012-2014). (c) Calibration (dashed line) and validation (solid line) of control model against SOC measurements (circles) taken in 2013,
2015, 2020. (d) Measured (unfilled circles), simulated SOC (grey solid line) and SOC with biochar-C (black solid line) in model validation. Biochar-C (filled circle) and simulated biochar-C (dashed line) represent the amount of biochar in soil without native SOC. Month of simulation 0 corresponds to March 2012, month of simulation 105 corresponds to December 2020.



## 4 Discussions

### 4.1 Model performance

The RothC model successfully simulated the SOC trend over time in both control and biochar treated soil. However, even after calibration, the control model overestimated the SOC in the first year. This could be due to the methodological difficulties in measuring SOC changes at yearly scale, because of the high spatial variability of soil in terms of SOC content (Hoosbeek et al. 2004; Coleman et al. 2004). Furthermore, the estimation of SOC is affected by the soil bulk density, which

can vary over time. In fact, the bulk density in the first year was decreased by soil disturbance during hoeing and digging performed in 2012 during biochar application, and in the control plots to ensure homogeneity; this could have led to an underestimation of the SOC stock in early 2013. Therefore, SOC simulations could better represent the trend of SOC over time in comparison to measurements. The same problem was not noticed for biochar stock, as it was determined using isotopic techniques, which allowed to estimate the biochar-C in the sampled soil layers more precisely and to distinguish it

from SOC, independently on bulk density change.

However, the cause of the decrease in SOC in time observed both in the control and in the biochar plots remains unclear. A possible reason could be that the previous soil management led to an increase in SOC to values larger than those expected for a soil in equilibrium with the current management condition (i.e., the poplar short rotation coppice plantation). Historical field data would be needed to confirm this hypothesis.

The simulation of the soil $CO_2$ flux from the RothC control model showed a marked difference with soil respiration data only during summers 2012 and 2013. We can hypothesise that the reason for this difference is the response of the SOC degradation function to soil moisture. During the summer periods of 2012 and 2013, the soil in the study area was rather dry, reaching values of 0.1 $m^3\,m^{-3}$ in July 2012 and 0.08 $m^3\,m^{-3}$ in September 2013 (Ventura et al. 2019a). Due to the soil water content decrease in summer, RothC predicted a decrease in SOC mineralization rates in summer. However, no such effect of

soil moisture changes on soil respiration was observed in the field. The overestimation of the effect of soil moisture deficit on soil respiration is likely due to the fact that its empirical equations have been calibrated in different climatic and soil conditions. It is possible that the soil microbial community in the study area is adapted to relatively dry summer conditions. The adaptation capacity of soil microbial community has been previously reported (Brangarí et al. 2020; Shu et al. 2021). Todman and Neal (2021) reported that drying and rewetting events can modulate soil microbial dynamics, inducing long

lasting microbial acclimation or adaptation responses. For this reason, we hypothesise that a modification of the empirical equation between matric potential and the effect on soil respiration by adapting it to the specific site could improve the fitness of the simulated $CO_2$ curve to the experimental data.

The heterotrophic respiration simulated in this study using RothC showed a better data fit than that of the ECOSSE model (Dondini et al. 2017), which was evaluated using experimental data collected between 2012 and 2014 in the three different




sites, including the same dataset used in the present study. In fact, the ECOSSE model systematically underestimates the
       $CO_2$ flux, particularly in the summer period when the simulated values are about half the observed values (Dondini et al.
       2017).

       The biochar model validation was also satisfactory, and this seems to support the assumptions made on biochar-C dynamics
       in soils: (a) biochar-C may not be directly mineralized to $CO_2$ but may partly transfer to pools with different degradation
rates (BIO and HUM); (b) the dependence of the biochar degradation rates on environmental variables can be made explicit
       (i.e., by creating a relationship of biochar degradation rates from air temperature, soil moisture deficit and soil coverage). In
       fact, the inclusion of biochar in RothC allows to modify degradation rates using climatic parameters. According to the BC-
       RothC model prediction, the remaining biochar amount eight years after the start of the experiment is only about 5% lower
       than the measured value.

**4.2 Long term biochar degradation and priming effect on SOM**

       Biochar degradation was faster than expected on the basis of previous studies in the same experimental conditions (Ventura
       et al. 2019a). Using a double exponential decay model, Ventura et al. (2019a) predicted that 70% of the initial biochar
       amount would still be present in the soil after eight years. According to this model, the values of $k_{LAB}$ and $k_{REC}$ obtained by
       the authors (2.55 and 0.08, respectively), which does not take environmental variables into account, were smaller than those
obtained in the present study.

       This confirms the importance of calibrating the models on the base of the specific environmental conditions. The use of
       empirical models outside of the specific conditions in which their parameters were determined can lead to the inability to
       evaluate the reliability of the model results and projections. In this case, this would mean using empirical parameters
       determined in laboratory experiments to simulate field conditions, or determined on the short term for projection to the long
term. Compared to the degradation rates found by Wang et al. (2016) in their metanalysis ($k_{LAB}$= 0.29 $y^{-1}$ and $k_{REC}$= 0.0018
       $y^{-1}$) which included mainly laboratory incubation studies on different types of biochar, $k_{LAB}$ is one order of magnitude lower
       and $k_{REC}$ is two orders of magnitude lower than those observed in this study. This highlights the importance of field studies
       to assess biochar degradation in soil, as a number of factors, such as climatic and environmental conditions, can increase its
       degradation rates in comparison to laboratory conditions.

Nevertheless, biochar showed a sufficient stability in soil to represent a valid strategy to increase soil C sequestration. In
       fact, in the evaluation of biochar degradation rate, it is fundamental to account for the original organic material, since
       different residues (e.g., leaves vs wood) already possess very different initial mineralization rates without charring (Lehman
       and Joseph, 2015). The biochar used in this study derives from maize silage, so we should assess its degradation rate in
       comparison with its original feedstock, not with other types of biochar.

In a study on biochar modelling, Lefebvre et al. (2020a) investigated the degradation of SOC under three scenarios, by
       considering different possible positive priming effects (0%, +21% and +91%) of biochar on SOM. Those values were taken
       from previous studies (Wang et al. 2016; Zimmerman and Ouyang 2019) and they are not specific to the biochar and soil



used in the experiment. On the contrary, we included in the modified RothC model only the negative priming effect observed in the field study by Ventura et al. (2015b) (-16%). This result is consistent with similar studies on maize biochar

degradation in soils, which generally report an initial release of $CO_2$ flux from soil, followed by a decrease in soil respiration, due to the negative priming effect of biochar on SOC mineralization (Luo et al. 2011; Zimmerman et al. 2011b). Since the validation of the biochar model against SOC showed very good results, we can conclude that the implementation of the priming effect in the RothC model is correct, at least for the specific experimental conditions of this study. This confirmed the persistence in the long term of the negative priming effect on SOC mineralization observed by Ventura et al. (2015b) in a

three years field experiment and by Stewart et al. (2013) in a two years incubation trial; however, the underlying mechanisms remain uncertain. A simple sensitivity analysis carried out on the BC-RothC model showed that the most important parameter in determining soil carbon sequestration potential of biochar is the priming effect factor; as such, more research in understanding and modelling the underlying processes is of paramount importance (see supplementary materials).

Different mechanisms have been proposed by Ventura et al. (2015b) to explain negative priming effect of biochar on native SOC, such as: (a) dilution effect of the soil microbial biomass; (b) OM sorption to biochar and consequent physical protection from degradation, particularly relevant in the short term (Zimmerman et al. 2011c) ; (c) substrate switching due to the preferential utilisation, by soil microorganisms, of the more easily available C represented by the labile biochar fraction (Stockmann et al. 2013; Abbruzzini et al. 2017). Using the results for the long term degradation of biochar we can elaborate

further on this problem; as it was done by Jiang et al. (2019). In this case, the effect of OM sorption to biochar should show a decrease in priming effect in time due to saturation of the adsorption sites; however, the measurements show no change in the priming effect. Moreover, on the long term, the labile fraction of biochar is already mineralized, hence it cannot explain the priming effect observed in this study.

It is known that biochar added to soils can induce changes in microbial communities, but the nature and the extent of those

alterations are poorly understood (Jenkins et al. 2017). Figures 2a and 2b suggest that the heterotrophic soil respiration in the control and in the biochar treated plots behave in the same way with respect to seasonal changes in soil moisture and soil temperature. This suggests that the activity of the microbial population in the soil was not affected by the application of biochar. Furthermore, the presence of the biochar in the soil did not result in any consistent change in soil moisture or temperature in the soil (Ventura et al. 2015b; Ventura et al. 2019a). This suggests that, in this study, the behaviour of the

microbial community of the soil was not modified by the addition of biochar.

### 4.3 Comparison with other biochar models

To the best of our knowledge, this is the first time that a simulation model, optimized for the prediction of biochar mineralization and C sequestration potential, has been calibrated and validated with long term field data.

Mondini et al. (2017b) suggested a method to include different types of soil amendments into RothC, including green waste

biochar. The RothC biochar modified by Mondini introduces two additional C-pools in RothC, one for resistant and one for



decomposable Exogenous Organic Matter (EOM, i.e. amendments), with degradation constants that vary depending on the type of amendment. One of the possible EOM is biochar; however, it was not possible to estimate its degradation constants due to very low values in measured $CO_2$ from mineralization. Furthermore, Mondini et al. (2017b) assumed that EOM added to the soil does not alter the degradation of SOC and excluded mechanisms of SOC stabilization/destabilization by biochar,

such as the priming effect.

Lefebvre et al. (2020a) modified the RothC model to evaluate the C sequestration potential of biochar from sugarcane fields upscaling the results to the whole São Paolo State, in Brazil. The degradation rate of the recalcitrant biochar fraction was estimated from one year incubation study with sugarcane biochar (Zimmerman et al. 2011c). The authors assumed, however, that the mineralization of the labile fraction of biochar can be simulated using the degradation rates of DPM and the RPM

pools. By contrast, in the present study the BC-RothC model was modified and parametrized based on the results of a three year experiment (Ventura et al. 2019a) and validated in the long term (eight years). Furthermore, Lefebvre et al. (2020a) evaluated their model with literature data from continuous addition of rice straw biochar to wheat maize cultivation (Liu et al. 2020), but not calibrated nor validated for the specific experimental conditions.

The results of this study are valid in the specific experimental conditions (soil type, climate, biochar type); therefore, they

should be applied with care to conditions that departs significantly from those of the experiment. We have introduced in the model what we have observed, for example the negative priming effect, and not the processes underlining the observations (i.e., the process leading to the priming effect). Therefore, we cannot exclude that different biochar degradation rates, or interactions with SOC could be observed in other conditions, for example a positive priming effect, or biochar leaching to the deeper soil profile. It should be taken into account that, in the literature, it is not clear yet how the priming effect process

works, and which environmental variables determine it. Further research is needed to understand the priming effect causes and mechanisms.

## 5 Conclusions

The understanding and assessment of the C sequestration potential of biochar requires the development of models able to take into account the turnover of biochar-C and SOC and effects on the SOC of added biochar. This study shows that our

modification of the RothC model was successful in simulating the dynamics of SOC and biochar degradation in soils in field conditions. As far as we know, this is the first soil C dynamic model including biochar that was calibrated and validated with long term field data. Results of the modelling and experimental measurements showed that, under the observed conditions, maize biochar degrades at much faster rates than in laboratory incubations or short term trials, and that biochar reduces the degradation of SOC. These results substantially confirm the findings of previous studies performed in the same site in the

medium term, remarking the importance of long term field studies to validate the results obtained in laboratory experiments. Nevertheless, biochar contributed substantially to increase the soil C stock in the long term, confirming its potential as a strategy to mitigate climate change.





*Acknowledgments.* This work was performed under the support of the 7th Framework Programme for Research and
Technological Development (FP7) of the European Commission (EUROCHAR project, N 265179). We kindly thank Fabio
Petrella and the I.P.L.A. (Istituto per le Piante da Legno e l'Ambiente) for providing the experimental site and for the help
during its establishment and maintenance.

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
