# Peer review of "Inclusion of biochar in a C-dynamics model based on observations from a 8 years field experiment"

_SOIL, 2021_

## Author Response (AR1)

**Answer to reviewers, manuscript "Inclusion of biochar in a C-dynamics model based on observations from a 8 years field experiment" by Roberta Pulcher, Enrico Balugani, Maurizio Ventura, Nicolas Greggio, Diego Marazza.**

We the Author want to thank the two reviewers for the insightful comments and suggestions. Here below I report the point-by-point reply to the comments, divided by reviewer, showing the original comments in blue and our answers in black.

*Reviewer #1 (Hans-Peter Schmidt)*

To the reviewer's knowledge, the paper presents for the first time biochar degradation data from a long-term field trial (8 years). It uses the data to modify the critical RothCmodel for soil C fluxes. The paper is well written in good English, and the results are well documented and analyzed.

The reviewer has only one major critic and suggestion: The biochar used in the experiment is a very particular biochar used in many European projects but is not truly representative of biochar on the European and global market. The biochar was made with a gasifier at HTT around 1200°C, while most biochars are made between 450 and 750°C. Despite the high HTT, the H/C ratio is 0.5 and thus very high, likely indicating the adsorption of labile carbon from pyrolytic gases. Moreover, the biochar is very brittle and has small particles sizes.

Dear Reviewer, first of all, the authors want to thank you for your nice words and the insightful comment. The authors are aware that the type of biochar is one that degrades relatively quickly, and as such we included the full information on its properties, in this case, specifically the H/C ratio of 0.5. The fact that this biochar is not representative of the materials used in the European biochar market was pointed out explicitly in the discussion, section 4.2, lines 392-395.

Discussing these particular properties of the biochar is very important in the context of the paper. Because losing 40% of the biochar carbon through biological or chemical degradation in only eight years is very unlikely and should not be hypothesized without measuring gaseous losses from the system. It is more likely that the friable, fine-grained biochar leached from the upper soil (only the top 40 cm were investigated) in the form of dissolved organic carbon. The latter cannot be claimed with certainty either. However, it should be discussed as the implication for Carbon-sink certification are enormous.

Unfortunately, the reviewer is not sufficiently familiar with the RothC-model and cannot judge with certainty if the modification of the model was done correctly.

We included in the manuscript an extensive discussion on possible unaccounted causes for biochar decrease in the soil (section 4.2, lines 365-386). There, we not only reported all the points discussed here below about leaching, but also addressed the problem of soil erosion.

1. The Biochar left in the field after 8 years is anyway much lower than that expected from the literature for a H/C 0.5 Biochar.
2. The H/C 0.5 Biochar degradation time estimates in the literature come from laboratory analysis, while our results come from the field – as such, other processes that could increase the speed of mineralization of the biochar cannot be ruled out, as pointed out in section 4.2. We are working in the direction of better understanding the biochar mineralization processes in the field that are not yet accounted for, and to do that, we are actually implementing the Long Term Experiment Platform to test and validate processes assumptions (https://site.unibo.it/environmental-management-research-group/en/long-term-platform ).
3. The biochar mineralization calculated after 8 years is in line with the mineralization rate estimated after 3 years, 3 years during which gas chambers were operated in the site and, thus, the gaseous losses were measured continuously.
4. Biochar leaching cannot be ruled out completely, however it is not very likely, since our measurements over the soil profile did not show an increase in SOC or TOC in the deeper (20-40

cm) profile in time. The soil texture was very fine, and this makes the leaching less likely. We are actually designing an experiment to verify whether there was any biochar leaching in the soil. Finally, we recognize that this is something that our study is not accounting implicitly, and as such is identified as a study limitation in section 4.3, line numbers 420-425.

*Reviewer #2 (Gabriel Sigmund)*

The authors present a very nice and clear study that is well structured and easy to follow. The results and the discussion thereof are well supported by the field data and the selected modeling approach. I have only one major comment, and a couple of minor suggestions below, before I can recommend publication.

Main comment:

The potential of downward migration of OM from SOM as well as biochar, including the potential colloidal transport of biochar (nano)particles is an aspect often neglected in studies on BC stability. Although the authors mention this aspect, I believe the discussion of this factor and the uncertainties associated with our understanding of these processes should be expanded on. In addition, the authors may consider to also further elaborate on the (potential) aggregation of biochar particles with clay minerals, which could have an important additional stabilizing effect, and may explain some of the variation when comparing BC stability in different soil types.

We must admit we did not measure leaching directly in the field experiment. However, during the first three years of the experiment, we measured both soil respiration and soil content of both naturally occurring SOM and added biochar. The results of the model, together with these measurements, point to the fact that if there was any leaching, it was negligible – at least in the first 3 years of experiment. The soil of the study area has texture 12% clay, 34% silt, e 54% sand, very similar (but with less sand) to a sandy loam soil used in a recent columnar experiment (Schiedung 2020); in that study, only 1% of the biochar was reported as being leached, most of it leached during the first flushing of the column. Finally, we measured the biochar in the soil at different depths, and the concentration of biochar at different depths did not change in time, i.e. we did not measure any downward migration of the biochar in the soil profile.

What happened after the first 3 years of experiment, however, is more debatable, since no soil respiration measurements were available for years 4 to 8. However, we believe the situation would not change much, since the biochar was already aged during the first 3 years.

We included this discussion, together with the response to reviewer #1, on lines 364-486.

We also discussed the interaction of biochar and clay particles observed visually (but not measured), on lines 387-391.

Minor suggestions

Line 74 delete „the" in „biochar submodel fort he RothC" or add „model"

Fixed

Line 191, 202 and subsequent occurrences: substitute "in the same site" with "for the same site" or "at the same site"

Fixed in all occurrences

Line 132 "via" the meteorological station

Fixed

Line 397 and 400 I would be a little more careful in the statement on BC effects on soil (micro)biota; perhaps elaborate to "not substantially affected"?

We clarified that we were referring to the change in microbial activity rather than changes in the microbial population: since the response of soil respiration to changes in temperature and soil moisture show no difference in treated and untreated plots, we can conclude that the *activity* of the microbial community did not change substantially (see lines 426-428).

---

## Author Response (AR2)

**Point-by-point response to Topical Editor Gabriel Sigmund**

Dear Gabriel, thank you for your comments. We modified the manuscript according to them:

1. The reference to the platform was "formalized" on lines 379-380 of the markup version of the manuscript, and we moved the reference and the link to the reference section as required.
2. We rephrased the sentence (now on lines 384-385) as requested, including three new references (added as well in the reference section).
3. We also found the sentence lacking in focus and clarity. Thus, we clarified half of the paragraph, now on lines 391-395, hopefully we were able to improve it.

Kind regards,

Enrico Balugani, on behalf of the Authors.